# Nitric Oxide in Cardiac Surgery: A Review Article

**DOI:** 10.3390/biomedicines11041085

**Published:** 2023-04-03

**Authors:** Nikolay O. Kamenshchikov, Nicolette Duong, Lorenzo Berra

**Affiliations:** 1Cardiology Research Institute, Tomsk National Research Medical Center, Russian Academy of Sciences, 634012 Tomsk, Russia; 2Department of Anaesthesia, Critical Care and Pain Medicine, Massachusetts General Hospital, Boston, MA 02114, USA; 3Department of Anaesthesia, Harvard Medical School, Boston, MA 02115, USA; 4Respiratory Care Service, Patient Care Services, Massachusetts General Hospital, Boston, MA 02114, USA

**Keywords:** nitric oxide, cardiac surgery, myocardial and acute kidney injury

## Abstract

Perioperative organ injury remains a medical, social and economic problem in cardiac surgery. Patients with postoperative organ dysfunction have increases in morbidity, length of stay, long-term mortality, treatment costs and rehabilitation time. Currently, there are no pharmaceutical technologies or non-pharmacological interventions that can mitigate the continuum of multiple organ dysfunction and improve the outcomes of cardiac surgery. It is essential to identify agents that trigger or mediate an organ-protective phenotype during cardiac surgery. The authors highlight nitric oxide (NO) ability to act as an agent for perioperative protection of organs and tissues, especially in the heart–kidney axis. NO has been delivered in clinical practice at an acceptable cost, and the side effects of its use are known, predictable, reversible and relatively rare. This review presents basic data, physiological research and literature on the clinical application of NO in cardiac surgery. Results support the use of NO as a safe and promising approach in perioperative patient management. Further clinical research is required to define the role of NO as an adjunct therapy that can improve outcomes in cardiac surgery. Clinicians also have to identify cohorts of responders for perioperative NO therapy and the optimal modes for this technology.

## 1. Nitric Oxide and Organ Protection in Cardiac Surgery

The literature indicates that cardioprotective effects are seen in the implementation of the early preconditioning (PC) phase and require multifactor adaptive signals, including physical, chemical and metabolic signals, such as NO. A longer during of PC additionally has a positive effect on myocardial stunning. As endothelial NOS (eNOS) is a mediator of late PC, the delayed cardioprotective properties induce eNOS activity and enhances tolerance to ischemia. Early and late PC can be triggered by the same stimuli; however, their results are fundamentally different in duration, signaling, physiologic and clinical effects. Clinically, nitric oxide’s (NO) application is seen in molecular and clinical studies: NO donors can reduce reperfusion injuries, and inhaled nitric oxide (iNO) administration correlates with lower rehospitalizations. The benefits of NO administration can also be seen in restoring blood flow, providing I/R cell protection and reducing inflammatory markers in organ injury.

### 1.1. Cardiac Surgery Risk Factors

The perioperative period of cardiac surgery is associated with damaging factors related directly to surgical aggression, cardiopulmonary bypass (CPB), bleeding, hemodynamic instability and cardiac arrest under conditions of protected anoxia. Cardiac surgery is followed by appropriate changes in neural reflex regulation, humoral activity and metabolic status [1]. These changes depend on initial the patient condition, surgery and CPB [2]. Acute myocardial and acute kidney injuries (AKIs) are among the most frequent and serious complications of cardiac surgery with CPB, determine the manifestation of multiple organ failure and have a significant negative impact on the overall prognosis in this patient population. Perioperative organ injury is independently associated with an increase in the cost of treatment and subsequent rehabilitation of patients after cardiac surgery [3]. It is extremely important that these effects can also occur in “subclinical” courses of organ dysfunction that remain undiagnosed. Despite the myocardial cardioplegic protection, clamping of the aorta during CPB leads to ischemia/reperfusion (I/R) injury. I/R injury of the myocardium is an important mechanism of myocardial dysfunction. A damaged myocardium can result from cardiomyocyte injury, tissue dissection and cardioversion [2]. The endothelial dysfunction that develops during I/R and the direct cytotoxic effects of neutrophils with microcirculatory distress and inflammation compartmentalization in the myocardium play an important role in pathogenesis [4]. Clinical manifestations of I/R injury in cardiomyocytes vary from transient myocardial dysfunction to myocardial infarction. Myocardial stunning is a reversible post-ischemic myocardial dysfunction in the presence of adequate restored coronary blood flow with no myocardial necrosis. Even with adequate perioperative cardioplegic cardiac protection, it causes low cardiac output and is characterized by impaired cardiomyocyte function [5]. Inadequate protection of the myocardium remains the main cause of complications during surgery with CPB [6]. Increases in the overall morbidity, in-hospital stay and long-term mortality are observed in patients with myocardial damage [7,8,9,10,11]. In general, perioperative mortality rates with these interventions range from 2 to 10%, depending on the left ventricle dysfunction severity [12,13,14]. Perioperative myocardial injury occurs in 3–30% of cardiac surgeries and is the leading cause of mortality [15]. Postcardiotomy cardiogenic shock during cardiac surgeries ranges from 2 to 6% [16]. Approximately 40% of cardiac patients with cardiogenic shock have right ventricular dysfunction [17]. As cardiac surgery is developing, the search for optimal cardioprotective approaches and techniques is still occurring.

Cardiac surgery-associated acute kidney injury (CSA-AKI) is a common and serious complication associated with increased morbidity and mortality. CSA-AKI is multifactorial and includes interrelated mechanisms: a neuroendocrine response to surgical stress, I/R injury, systemic and organ inflammation, oxidative stress, CPB-induced intravascular hemolysis and microembolization [18]. The development of AKI in intensive care unit (ICU) patients promotes an increase in the frequency of infectious complications, length of stay, hospital re-admission and 30- and 90-day mortality [19]. Long-term outcomes for AKI demonstrate an increase in the development of AKD, chronic kidney disease (CKD), long-term risk of graft loss, dialysis dependence and mortality [20]. Patients with a history of AKI have an increased rate of cardiovascular and cerebrovascular complications [21]. CKD is associated with an increased risk of cardiovascular diseases [22]. Conversely, CKD affects the initiation and/or progression of renal dysfunction [23]. These facts indicate the pathogenetic mechanisms underlying both heart failure and AKI or renal failure. In this regard, organoprotective strategies for perioperative protection of the heart and kidneys may be applicable.

This review presents a state-of-the-art view of the possibilities of perioperative nitric oxide therapy for adjuvant organ protection in cardiac surgery. NO acts as a key preventative of cardiorenal syndrome [23]. In clinical practice, heart and kidney failures are recognized as comorbid pathologies; dysfunction in one organ can accelerate the course of pathological changes. NO is critical in the understanding and clinical implementation of the pathogenetic mechanisms of multimodal perioperative organ protection (Figure 1). Moreover, the strategy of NO-mediated protection can reduce multiorgan complications outside the cardiorenal axis, such as injuries to the lungs and visceral organs.

The accumulated data confirm the central role of mitochondria in the implementation of the protective effects of NO in IR injury. NO-dependent activation of cGMP-protein kinase G (PKG), as well as PKG activation outside the cGMP-dependent pathway, mediates the opening of mitochondrial KATP channels. Opening the KATP channels reduces Ca^2+^ overload, and the associated intramitochondrial signaling leads to deactivation of the mitochondrial permeability transition pore (mPTP), resulting in a cytoprotective phenotype. [24]. NO is able to act outside of these mechanisms, due to its binding to the active centers of cytochrome c oxidase, which inhibits the apoptotic signal from mitochondria. [25] Due to post-transcriptional modifications, NO modulates inflammatory responses, including NF-κB pathway signaling, and also decreases reactive oxygen species (ROS) generation and acts as an antioxidant in ischemia-related injury. [26] Another target of NO-therapy may be the endothelial protective effect against platelet aggregation and leukocyte adhesion, a reduction in damage to the system of microcirculation and optimization of substrate delivery and provision of tissues [27,28].

### 1.2. Biological Significance of Nitric Oxide 

Nitric oxide (II) or nitric monoxide is a colorless gas. The structural formula is N=O. The understanding of this molecule has become one of the greatest achievements of the 20th century. The 1992 Science journal called nitric oxide or the accepted abbreviation “NO” the molecule of the year [29]. In 1998, Drs. Ferchgott, Murad, and Ignarro were awarded with the Nobel Prize for the discovery of the NO’s role as a signaling molecule in the cardiovascular system [30].

In human organisms, nitric oxide NO is formed in the presence of nitric oxide synthase (NOS). The substrate for the synthesis is L-arginine, the oxidation of which to L-citrulline and NO is formed by NO synthases [31]. Three NOS isoforms are present: endothelial nitric oxide synthase (eNOS or NOS3, located in the vascular endothelium), neuronal nitric oxide synthase (nNOS or NOS1, located in neurons), and inducible nitric oxide synthase (iNOS or NOS2, located in the immune system) [32]. 

Until 1970, NO was considered a toxic environmental pollutant, the production of which could be seen from an internal combustion engine emission into the environment [33]. The cytotoxic properties of NO create the innate immune system’s first line of defense against foreign invaders. Tissue macrophages realize their effects through the peroxynitrite-dependent mechanism of cytotoxicity with the synthesis of superoxide anion radical and NO [34]. Peroxynitrite, seen with phagosomes, facilitates pathogen destruction. The first mechanisms of peroxynitrite and NO cytotoxic effects consist of the direct oxidation of biological metalloproteins [34,35,36,37]. These molecules can then damage biological targets via the free radical mechanism [35]. However, the spectrum of NO-regulatory effects expands beyond its cytotoxic effects. NO has enormous diffusion ability and is phylogenetically adapted to participate in signaling cascades as a universal transmitter [38]. The covalent modification of proteins and nucleic acids mediated by NO and peroxynitrite can lead to aberrations in enzyme activity, changes in intra- and cross-cellular signaling and transformation of the cell ultrastructure [39]. 

The interest in local blood flow autoregulation arose with the discovery that endothelial cells of blood vessels emit a factor capable of causing blood vessel relaxation [40]. Endothelium-derived relaxing factor (EDRF) was additionally discovered to inhibit platelet aggregation [41]. Further study established that EDRF releases and exhibits the pharmacological, biochemical and chemical properties of NO [42,43,44]. The main physiological effect of the increase in the cyclic guanosine monophosphate (cGMP) concentration is the relaxation of the coronary artery muscular layer [45]. The implementation of NO’s physiological effects occurs through the activation of the effector enzyme-soluble guanylate cyclase inside the smooth muscle cells of the vascular wall, which catalyzes the second cGMP messenger [41,46]. The attachment of NO to the heme prosthetic group, with the formation of a nitrosyl complex and conformational changes in the enzyme, leads to a several-fold increase in the activity of the guanylate cyclase-cGMP system [47]. 

The NO-dependent mechanism of transmembrane signal translation does not require G-protein mediation due to the high diffusivity of NO. The increase in the cGMP concentration activates cGMP-dependent protein kinase G (PKG) and leads to two effects [48]. The former effect phosphorylates phospholipase C and causes the repression of inositol triphosphate formation. The latter releases Ca^2+^ from the sarcoplasmic reticulum to the cytoplasm [48]. At the same time, phospholamban phosphorylation has an inhibitory effect on Ca^2+^-adenosine triphosphate (ATP), leading to an increase in Ca^2+^ transport from the cytoplasm to the sarcoplasmic reticulum cisterns [49]. The cumulative result of these reactions within these two processes is a reduction in the Ca^2+^ level in the smooth muscle cell cytoplasm. Thus, the activity of the Ca^2+^-dependent kinase of the light chains of myosin decreases, and the result is smooth muscle relaxation [50]. 

### 1.3. Current Paradigm of Clinical Applications and the Nitric Oxide Therapeutic Niche

NO is a drug that causes the relaxation of smooth muscle cells in the vasculature and selective pulmonary vasodilation when administered via inhalation. At the same time, the molecule has a short half-life, does not enter the systemic circulation and does not cause systemic effects on blood pressure. However, the possible implementation of extrapulmonary effects of NO is associated with the accumulation of its metabolites in distant organs, which will be discussed later. [51]. iNO is currently used for the treatment of pulmonary hypertension (PH) and is also used as a rescue therapy in patients with acute respiratory distress syndrome (ARDS) [52]. ARDS is characterized by increased intrapulmonary shunting of blood through hypoventilated regions and an increase in pulmonary vascular resistance (PVR), with 21% of ARDS cases occurring when patients develop acute RV failure [53]. iNO, reducing pulmonary vascular resistance, may improve VPR and relieve hypoxemia in ARDS [54]. Despite improved oxygenation, iNO does not reduce mortality and may even be harmful due to an increased incidence of renal complications. [55]. Similar data were obtained from COVID-19-associated ARDS [56]. Current guidelines recommend against the routine use of iNO in mechanically ventilated patients [57]. In contrast, iNO is widely used in PH. In persistent pulmonary hypertension of the newborn, iNO improves arterial oxygenation and systemic hemodynamics and reduces the need for extracorporeal membrane oxygenation (ECMO) [58]. The use of iNO after congenital heart surgery can reduce PVR, lessen the risk of PH crises and shorten the postoperative course [59]. PH often increases morbidity and mortality in the general population of cardiac patients. iNO is frequently administered in these patients to prevent or reverse RV failure and cardiogenic shock, enhance RV output, reduce the risk of RV dysfunction and improve survival in patients with PH after a heart transplant [60]. 

The use of iNO in cardiac surgery leads to important physiological goals: selective pulmonary vasodilation reduces afterload for the right ventricle (RV) and optimizes performance and myocardial oxygen demand, exerting a cardioprotective effect (Figure 1). On the other hand, high venous pressure (i.e., right atrial or central pressure) in patients after heart surgery is associated with kidney function worsening. [61]. Thus, iNO can have a protective effect on renal function by reducing RV afterload, resulting in lower right-sided filling pressure and the prevention of renal venous congestion. [62] (Figure 1).

### 1.4. Nitric Oxide and Myocardial Protection from Ischemia–Reperfusion Injury

The clinical use of nitrates and attempts to interpret their biological significance have been undertaken since the mid-19th century, when the efficiency of this group of agents was first discovered in patients with angina [63]. Currently, NO donors are common drugs used to treat cardiovascular diseases. Researchers have shown a strong interest in the role NO in cardioprotection, and a comprehensive review from 2016 has been included to emphasize the role of I/R in preconditioning (PC) [64].

The role of NO in the modulating effects of I/R injury in the heart and in the preconditioned myocardium is fundamentally different. Moreover, the phenomenon of ischemic, pharmacological or remote preconditioning includes two-time phases of protective effects with different pathophysiological mechanisms in the implementation and clinical consequences. In this regard, the role of NO in the early and late phases of PC will be considered separately in the following sections [65,66]. 

#### 1.4.1. The Role of Nitric Oxide in the Early Phase of Preconditioning

Multiple research projects were devoted to NO-mediated cardioprotection in the unpreconditioned heart. Studies were based mainly on two methodologies: applying pharmacological inhibition of NO-synthase activity to reduce the endogenous NO pool before, during or after the ischemic event, as well as increasing the bioavailability of NO during I/R injury of the myocardium by introducing donors or precursors. 

Until recently, the controversial literature supported the detrimental effects that NO has on the heart due to the generation of free radicals and pro-oxidant action during myocardial ischemia. The largest analysis of the effect of NO on I/R injury of the myocardium was conducted by R. Bolli with over 100 papers on the subject [67]. Studies support the conclusion that the inhibition or ablation of NOS endogenous activity exacerbates I/R injury of the myocardium, while the delivery of exogenous NO has a protective effect [67].

PC is a protective mechanism of organs and tissues adaptation after sublethal impulses. The early phase of PC develops within minutes after a short-term damaging stimulus, such as ischemia, pharmacological agents or humoral stimuli. Studies on ischemia-induced effects of the early PC phase demonstrated that the inhibition of endogenous NO synthesis by NO-synthase antagonists does not affect the protective effects of ischemic PC against cell death, post-ischemic dysfunction and arrhythmias [68,69]. At the same time, there is opposite data showing that NOS ablation impedes the development of early PC [70,71]. The studies devoted to PC also gave controversial results, stating that the implementation of acetylcholine-induced and bradykinin-induced PC was not blocked by NOS inhibitors, which is contrary to studies that revealed the acetylcholine-induced blockade of PC by the endogenous NO ablation [72,73].

Despite the inconsistency of data on the role of endogenous NO in early PC development, pharmacological PC using NO donors is a promising field. Similar to PC, evidence indicates that exogenous NO simulation has the same protective effects on the myocardium [69]. Researchers also carried out studies on the efficiency of the NO supply to the CPB circuit to protect the myocardium from I/R injury in simulating acute myocardial infarction (MI) under normothermic CPB in an experiment [74]. Its principal difference with this study is the use of the native NO molecule, and not its precursors, to simulate cardioprotection. Results confirm the cardioprotective properties of NO in the simulation of I/R myocardial injury. The cytoprotective effects manifested in the reduction of the infarction zone. [74]

This review summarizes studies that indicate that the implementation of the early PC phase cardioprotective effects induced by ischemia does not necessarily involve endogenous NO. Similar to cytoprotection in PC, the phenomenon of myocardial NO protection is observed. Conclusions about the sufficiency are logical in light of the modern understanding of PC mechanisms. The early PC is a multifactor adaptive response and can be caused by physical, chemical and metabolic signals. During sublethal stress, PC is triggered and releases NO, adenosine, bradykinin, opioid agonists, reactive oxygen species (ROS) and catecholamines [75]. For these reasons, the elimination of NO from the set of signals initiating early PC is not enough to block protective phenotype development.

#### 1.4.2. The Role of Nitric Oxide in the Late Phase of Preconditioning

Early-PC phase effects are seen within 1–2 h, whereas ischemic PC induces a delayed cardioprotection phase 12–24 h after the damaging stimulus and provides myocardial protection for 72 h [76,77,78,79]. Due to the longer duration, the late PC phase has the potential to prevent myocardial stunning. The significance of the delayed cardioprotection phenomenon may have greater clinical significance than the early PC phase [80,81,82,83,84,85,86] and differences are noted in Table 1.

It is important to note the difference between the triggers of late PC and the mediators of late PC. Data suggest that the development of late PC requires the presence of both NO and O_2_. These facts indicate NO as a chemical signal as the heart undergoes a transition to the protective phenotype in response to ischemic damage or intense overstrain stress.

It is highly likely that eNOS acts as a source of NO generation, as a mediator of the late PC phase. This hypothesis is supported by the fact that the implementation of the late PC phenomenon is blocked by the administration of a non-selective NO-synthase inhibitor [87]. A sharp rise in calcium-dependent NOS is observed after exposure to the ischemic stimulus; however, this effect is offset by non-selective NOS inhibitors, which block late PC [88].

The isolated administration of NO donors without simulated ischemia causes a delayed cardioprotection effect similar to the effects of the late PC phase. The infarct-limiting effect, as well as a positive effect against myocardial stunning, is comparable to the effects of sublethal ischemic stimuli. These facts confirm not only the necessity but also the sufficiency of the NO presence to induce late PC [80,82,85,89].

The signaling pathway of NO-mediated induction of the late PC phase involves ROS, protein kinase C^ε^ (PKC^ε^), tyrosine-dependent protein kinases, activated B cells and iNOS gene activation [85,90,91,92,93,94,95]. The upregulation of iNOS expression in ischemic PC is possible through the activation of the tyrosine kinase JAK1/2 and transcriptional activation factor STAT1/3 signaling pathway [96]. Notably, the activation of the same signaling pathway leads to the realization of the ischemic PC protective effects [67]. 

iNOS is responsible for NO generation in late PC. Studies convincingly show that ischemic PC induces the expression of iNOS proteins, increases its activity within 24 h after the stimulus and leads to a complete leveling of the late PC cardioprotective effects [97]. iNOS is a mediator of several forms of pharmacological PC, including adenosine, opioids, NO donors and endotoxin derivatives [98,99,100,101,102,103,104,105]. 

Delayed iNOS-mediated cytoprotection leads to the participation of two different NOS isoforms. The conceptual basis is that the heart has a feedback mechanism by which NO produced by eNOS can regulate iNOS. It is postulated that NO generated by eNOS induces the late PC phase on the first day and further generation activates iNOS protection against ischemia over the next 24–72 h [98,105,106,107,108]. The upsurge in NO production during the PC stimulus has a protective role and causes the increased expression and activity of iNOS after 24 h. Currently, the NO hypothesis of late PC has been convincingly confirmed based on pharmacological, biochemical and molecular genetic data accumulated using various models of cardiac adaptation.

The late-PC NO hypothesis provides insight into NOS isoforms. eNOS is capable of being activated by Ca^2+^-dependent and Ca^2+^-independent mechanisms, which are applicable for an immediate but short-term adaptive response [109]. On the contrary, iNOS expression is inducible with activation by stress and does not require Ca^2+^ levels or cellular homeostasis [109]. This allows iNOS to be suitable for a delayed but prolonged adaptive response. When a damaging stimulus affects the unpreconditioned myocardium, eNOS mobilizes an immediate upsurge in NO biosynthesis to overcome the acute phase of crisis [110]. At the same time, a protective phenotype is being implemented to counter the possible repetition of aggressive patterns. This requires an increased level of myocardial NO for a sufficiently long period, which is supported by the activation of iNOS, and mediates the cardioprotective effect in cardiac adaptation and late PC [110]. The result represents a generalized mechanism of protection against ischemic injury. As an adaptive response, the expression of iNOS increases in response to hypoxia in cardiomyocytes in children with cyanotic congenital heart defects [111,112,113]. The heart responds to stress in a biphasic way, using eNOS as an immediate, and short-term protection and iNOS as a delayed, but long-term, adaptation.

Many hypotheses were proposed to explain the cardioprotective effect of NO in I/R injury, but the exact mechanism remains unclear [67] (Table 2). 

To date, the main pathway implementing the cardioprotective phenotype is considered to be the pathway of NO-dependent activation of soluble guanylate cyclase with the formation of cGMP, the end point of which is the activation of PKG [114,115]. At the last stage of cytosolic signaling, PKG affects the mitochondria, which leads to the opening of mitoKATP and inhibition of MPTP, which are molecular PC effectors leading to the protection of mitochondria in ischemia [116,117].

Another mechanism of myocardial protection against I/R injury was identified and consists of the NO-dependent stimulation of COX-2 activity, followed by the production of cytoprotective prostaglandins [118]. Pioneering studies revealed that the inhibition of iNOS activity in the effects of the late PC phase leads to a decrease in the synthesis of prostanoids [119]. These facts indicate that COX-2 activity is PC by the iNOS-generated NO, and the produced prostanoids are necessarily required for the cardioprotective effect [120].

Furthermore, NO administration can reduce the phenomenon of non-restored blood flow (“no-reflow”) [121], reduce leukocyte infiltration, the release of cytokines and expression of adhesion molecules, which prevents inflammation compartmentalization in the myocardium [122,123,124]. It is obvious that such pluripotent effects of NO outside the intramitochondrial signaling pathway potentiate its protective properties with respect to the myocardium.

The role of NO in the implementation of cardioprotective effects of the PC phenomenon is critically important. PC leads to eNOS phosphorylation, while the inhibition of this enzyme reduces the cardioprotective effects of the phenomenon [125,126]. Apart from the mediation of classical pre- and postconditioning, NO plays a pivotal role in cardioprotection induced by pharmacological agents. Thus, statin therapy leads to a change in the eNOS expression profile, which has a beneficial effect on both the course of the underlying cardiac disease and its tolerance to I/R injury in acute cases [127]. A number of studies confirmed the NO-mediated myocyte protective action in acute cardioprotection after a single pretreatment with statins; there are also reports of potential protective effects of statins during reperfusion [128,129,130]. iNOS induction is the central mechanism for the implementation of phosphodiesterase inhibitor protective effects [131]. Phosphodiesterase type 5 inhibitors show potent cardioprotective effects upon the simulation of I/R injury in animal models [132,133]. It is noteworthy that similar results were obtained in non-ischemic models of myocardial injury and were also implemented within the NO-dependent pathway [134,135].

iNOS-dependent NO biosynthesis is the ultimate common pathway by which the cardioprotective phenotype is implemented in stress [110]. Additionally, PC stimuli cause iNOS activation by an order of magnitude less than that observed after lipopolysaccharide infusion [97,136]. This explains the paradox of the iNOS cardioprotective effect: PC in cases of its moderate expression, and cytotoxic effects when it is hyperactive in cases of inflammation. Additionally, the activation of angiotensin type 1 receptors triggers specific cardiac effects associated with angiotensin-mediated superoxide production, which causes persistent oxidative stress and decreases the bioavailability of NO in cardiomyocytes [137]. The involvement of the NO pathway in the implementation of anesthetic PC of the myocardium as an option for pharmacological PC is actively discussed [4,138].

#### 1.4.3. Implementation of Nitric Oxide Cardioprotective Properties into Clinical Practice

The NO PC hypothesis has a therapeutic application that can be put into clinical practice. The discussed experimental studies showed that adjuvant therapy based on NO donors can simulate the molecular and functional aspects of ischemia-induced PC [80,82,85,89,92]. 

A 2016 review published clinical studies from 1990–2014, which used NO inhalation (iNO), nitrite, nitrate or NO donors to reduce reperfusion injury in experimental and clinical studies, upon which the value of these interventions in clinical myocardial infarction remained unproven [64]. Studies performed on patients who had percutaneous transluminal coronary angioplasty showed that the infusion of nitroglycerin protects the myocardium from ischemia 24 h after its administration [139,140]. The cardioprotective effects of intravenous sodium nitrite and iNO for ST-elevation myocardial infarction (STEMI) were additionally investigated in the NIAMI and NOMI studies [141,142]. Patients of NIAMI underwent magnetic resonance imaging (MRI), and no reliable evidence of the effectiveness of these techniques was obtained, in contrast to experimental studies, where the degree of myocardial damage was assessed based on cardiac markers [142,143,144]. In the NOMI trial, there was no significant difference between the primary (LV infarction size) and secondary endpoints in both groups, but it showed a tendency of the reduction of the infarction zone size in the risk zone and cardiac remodeling surrogates by MRI in the iNO group in 4 months. Upon death, recurrent ischemia, stroke or rehospitalizations showed a tendency toward lower event rates with iNO at 4 months and 1 year [141]. The results of these studies caused a great debate among scientists and physicians, which means it requires further iNO functions to reduce reperfusion injury in STEMI [145].

In addition, genetic and pharmacological strategies aimed at regulating a common mediator of protective iNOS using various types of PC seem promising. Experimental studies on eNOS or iNOS gene transfer demonstrated a powerful infarct-limiting effect [146,147]. Therefore, NOS gene therapy can be an effective strategy to achieve a local increase in the level of NO in the myocardium to combat I/R injury. This eliminates the need for the continuous administration of NO donors, and therefore, there are no systemic NO-dependent hemodynamic changes.

Cardiac surgery is a clinical model of heart injury mediated by I/R, which makes an important contribution to the pathogenesis of myocardial dysfunction in the postoperative period. Cardiac ischemia dramatically changes the metabolic profile and energy metabolism of cardiomyocytes, activates anaerobic glycolysis and contributes to a decrease in the synthesis of energy substrates in mitochondria. Global reperfusion injury may exceed the ischemic one in terms of severity [148]. 

The positive effect of glucose and lactate on metabolism was obtained, and the need for insulin and inotropic agents of nitroglycerin for intravenous administration started from the warming-up period during CPB and further in the ICU [149]. However, the use of NO precursors for cardioprotection may have certain limitations. In particular, the use of NO donors may lead to the development of hemodynamic instability and could be ineffective in conditions of acidosis and reductions in cofactors, such as ascorbic acid and polyphenols [150]. The natural production of NO is reduced during CPB, intraoperative replacement therapy with its physiological precursor L-arginine is almost impossible and intraoperative blood components of transfusion potentiate the sequestration of endogenous NO [151,152,153]. The intraoperative reduction in NO bioavailability causes systemic microcirculation disorders [154,155]. Persistent vasoconstriction due to NO deficiency can ultimately lead to organ dysfunction [156]. In this regard, the attention of many researchers was drawn to perioperative cardioprotection mediated by the native NO molecule. Despite the fact that I/R injury during cardiac surgery in adult patients is a problem, there are limited clinical studies confirming the cardioprotective effects of NO in this category of patients. Table 3 presents clinical studies of NO treatment for cardioprotection in different categories of patients in terms of cardiac surgery.

## 2. The Role of Nitric Oxide in Preventing Cardiac Surgery-Associated Acute Kidney Injury

Renal complications seen in cardiac surgery-associated acute kidney injury (CSA-AKI) can be addressed with affordable, functional procedures, such as remote ischemic preconditioning (RIPC). PC is another way to protect the kidneys by increasing NO release from NOS expression and triggering anti-apoptotic and anti-inflammatory effects. The PC pathways in the kidney and myocardium have similar functions. Likewise, the exogenous administration of NO can trigger protective measures in both systems. NO bioavailability, reduced intraoperatively, is associated with an increased risk of chronic kidney failure (CKF) and mortality. In cardiac surgery, hemolysis levels are associated with the prolonged use of cardio-pulmonary bypass (CPB). The administration of NO to the CPB circuit was seen to decrease AKI, severity and major adverse kidney events.

### 2.1. The Role of Nitric Oxide in Kidney Preconditioning

A high metabolic rate and vascular anatomy make the kidneys an extremely sensitive organ to I/R injury [166]. It is important to prevent renal complications using kidney protection technologies, emphasized earlier with PC. The essence of the PC phenomenon is short cycles of sublethal damage, which contribute to the formation of a renoprotective phenotype. In particular, remote ischemic preconditioning (RIPC) is a phenomenon whereby a brief period of I/R provides tolerance to subsequent periods of ischemia [167]. RIPC is also a common variant of induced kidney adaptation. It is triggered by brief episodes of transient ischemia and reperfusion, and the protective effects are not limited to the organ or tissue receiving the preconditioning stimulus but is also traceable in remote organs [168]. A number of clinical studies showed the efficiency of RIPC in preventing CSA-AKI, and RIPC is considered to be a preventive measure, because it is cheap, functional and not associated with complications [168,169]. One group focused on determining the optimal time for sublethal ischemic injury and found that post-ischemic proximal tubules were resistant to I/R and potentially protected against hypoxia, ROS and calcium ionophores [170,171,172].

The protective effect of various PC protocols on structural kidney damage was obtained in morphological studies on animals [173,174]. Experimental studies later confirmed the protective role of PC before prolonged ischemia in preserving renal function: a decrease in peak creatinine concentrations was observed from day 1 to day 7 [175,176]. PC additionally has a protective effect on the kidney after I/R injury since it reduces serum creatinine, blood urea nitrogen and renal damage [177].

Researchers examined the contribution of NO to the effect of PC on renal function and the hemodynamics in I/R-mediated kidney injury [178]. Results show that PC protective effects in relation to kidneys were least-wise partially mediated by NO. Further studies using eNOS-deficient and wild-type mice established the crucial role of constitutive endothelial nitric oxide synthase (eNOS) activity, and eNOS-mediated NO production plays a pivotal role in the renal-protective effect of PC against the I/R-induced AKI [179]. The protective effect of ischemic PC on I/R-induced AKI is closely related to the renal NO production following the increase in eNOS expression after reperfusion [180]. A significant increase in serum NO metabolites, eNOS, and iNOS expression were registered in another PC-group analysis. The authors concluded that PC has a protective effect on kidney structure and function, which may be produced by increased NO release arising from increased NOS expression. Animal kidney studies have shown that PC has acute and delayed phases of renal protection related to the increase in the expression and activity of iNOS after PC application [181].

There is evidence that NOS is an important target of ischemic adaptation in other organs [182,183]. In this context, the protective effect of NO in kidneys plays a key role in triggering PC phenomenon in different organs via its antioxidant, anti-apoptotic and anti-inflammatory properties [184,185,186]. Endoplasmic reticulum stress in renal cells was found to be critical in AKI in humans and animal models of I/R injury [187,188]. In pancreatic beta cells, the chronic inhibition of NO production exacerbates endoplasmic reticulum stress [189]. Early PC protects the kidney against kidney I/R injury by reducing oxidative and endoplasmic reticulum stresses [190]. 

The clinical application of PC organoprotective technologies can be in surgeries associated with kidney I/R. For cardiac surgery, it is impossible to implement RIPC of kidneys with the classic protocol of renal ischemia in cardiovascular surgery practice. However, one meta-analysis showed that RIPC leads to little difference in serum creatinine, adverse effects, the need for dialysis, length of hospital stay, death and AKI incidence. Concluding data did not confirm the efficacy of remote RIPC in reducing renal I/R injury in patients undergoing major cardiac and vascular surgery [168]. In contrast, another meta-analysis concluded that RIPC might be beneficial for the prevention of AKI following cardiac and vascular interventions, but the evidence is not robust enough to make a recommendation, and adequately powered trials are needed to provide more evidence in the future [191]. In any case, the issue of the PC and the RIPC in the prevention of CSA-AKI remains open.

Pharmacological agents that mediate protection using the signaling pathways of NO may lead to therapeutic improvements to reduce the incidence of AKI. These events are mainly accomplished by the eNOS pathway transduction [192]. This confirms that ROS and NO interactions are an important regulator of kidney function. An increased endogenous antioxidant system and reduced lipid peroxidation were introduced as probable NO-dependent mechanisms against kidney I/R injury [193]. Lithium pre-treatment may be promising in the setting of transplantation [194]. The results obtained by these authors indicate that lithium ameliorates kidney I/R injury through NO pathways. Another research study looked at renal I/R-induced functional and morphologic injury, which was significantly improved by prior treatment with resveratrol. It was established that resveratrol exerts its protective effect through the release of NO and also through its antioxidant mechanism [195]. Demonstrated treatments with erythropoietin or PC and their combination produced an improvement in tubular and glomerular function. These advantageous effects were closely related to reducing oxidative stress, suppressing proinflammatory response and enhancing the generation of NO via the upregulation of iNOS and eNOS gene expression [196]. Findings demonstrate that the upregulation of iNOS and eNOS is essential for erythropoietin and PC-mediated amelioration of kidney function in I/R injury. 

Post-CPB dexmedetomidine use is associated with a reduction in AKI incidence [197]. Subsequent studies demonstrated that cardiac surgery patients who received an intravenous dexmedetomidine infusion were more likely to have better in-hospital, 30-day and 1-year survival. The perioperative infusion of dexmedetomidine effectively reduced both the incidence and severity of AKI and improved outcomes in patients undergoing valvular heart surgery [198]. The renoprotective action of dexmedetomidine is pluripotent and is represented by a variety of different mechanisms, including activation of the NO signaling pathway, which is demonstrated by significantly increased eNOS expression under the influence of dexmedetomidine [199].

NO donors administered during the ischemic event appear to suppress tumor necrosis factor α (TNF-α) by modulating the activity of mitogen-activated protein kinases inclusively [200]. The expression of cytokines was found to be reduced by sodium nitroprusside after I/R injury [201]. NO has a demonstrated protective effect against TNF-α-mediated apoptosis and TNF-α-moderated cell death [202]. NO inhibits interleukin-1 release, which is a method of protection against I/R injury [203]. 

I/R injury leads to the upregulation of cell adhesion molecules (CAMs), which in turn leads to the activation of a variety of inflammatory molecules [204]. I/R decreases the NO level and increases CAM levels in the cell [205]. The exogenous NO donors used in I/R injury show anti-apoptotic effects by inhibiting TNF-α and reducing oxidative stress, serum or glucose deprivation [206]. The beneficial effects of an exogenous NO donor during an ischemic insult demonstrated that the downregulation of p53 caused by NO correlates with the apoptosis decrease [207]. 

The anti-apoptotic potential of NO may also be associated with the NO-induced expression of heat shock proteins possessing protective properties [208]. I/R injury induces the expression of macrophage inflammatory proteins contributing to the detrimental effects of reperfusion [209]. NO downregulates the production of these chemokines after I/R injury [210]. 

We can conclude that elevated levels of NO in tissues achieved by its exogenous delivery during I/R can lead to the activation of many pathways that may be able to change the expression of some genes that optimize tissue respiration. Together, with a decrease in the compartmentalization of inflammation in the organs, this can reduce apoptosis. Decreased apoptosis and cell inflammation can lead to an increase in organ protection, help to protect the cell during I/R and decrease organ injury.

Summarizing all of the above-mentioned data, this review concludes that the PC pathways in the kidney are identical to those in the myocardium. The underlying pathways activated in RIPC may be similar to those recruited in the setting of local PC. All aspects of the PC signaling pathway can be subject to pharmacologic manipulation and the development of pharmacological postconditioning strategies. Exogenous and endogenous NO serves as a trigger and mediator of protective phenotype implementation. The early PC phase realized by means of upregulation of the kinase cascade subsequently results in the activation of mitoKATP and inhibition of mPTP and lasts no longer than several hours [211]. The late phase starts 24 h after the preconditioning stimulus and can last for several days. The late PC phase depends on transcriptional responses that modulate glucose and mitochondrial metabolism, suppress ROS production and inflammation and inhibit the expression of proapoptotic molecules [212]. The second window of protection is mediated by the activation of nuclear transcription, which results in the production of protective mediators. 

### 2.2. Alternative Pathways and Specific Mechanisms of Nitric Oxide Cytoprotective Effects in Cardiac Surgery

Postperfusion syndrome in cardiac surgery under CPB includes the development of intravascular hemolysis [213,214]. The concentration of free hemoglobin in blood plasma increases five-fold from the initial level at the initiation of CPB, followed by a linear increase in extracellular hemoglobin [214]. In 25% of cases, increased cell-free hemoglobin (fHb) and reduced iron after I/R exceeds the serum iron-binding capacity and aggravates the reperfusion injury [215,216]. This phenomenon is associated with a decrease in organ perfusion and oxygen delivery to the kidney medulla; regional ischemia leads to tubular injury with aberrations in renal function in the postoperative period [217,218]. 

Intravascular hemolysis and hemoglobinemia induce the development of multi-organ injury with intravascular blood coagulation and the formation of kidney, respiratory and multiple organ failure [219,220,221]. Hemoglobinemia has a damaging effect on kidneys: the reabsorption of fHb leads to the formation of free iron and induces the generation of ROS and free radical damage [222]. Erythrocytes undergo hemolysis during the CPB and after transfusion. fHb scavenges endogenous NO leading to systemic and pulmonary vasoconstriction and the development of pulmonary and systemic hypertension [223]. 

In the postoperative period of cardiac surgery, systemic microcirculation disorders form the basis of multiorgan complications. The sequestration of NO is responsible for the relaxation mechanisms of the vascular wall mediated by fHb, being a pathogenetic mechanism. The reaction occurs quickly while deoxyhemoglobin binds to NO reversibly to form nitrosylhemoglobin, and OxyHb binds to NO irreversibly to form nitrate and methemoglobin (MetHb) [224].

fHb concentrations as low as 10 mg/dL can inhibit vasodilation in vivo [225,226,227]; however, depending on the phenotype of haptoglobin, the potential for binding of fHb and the antioxidant potential are sharply limited, due to the high prevalence of the Hp2-2 allele in the adult population of cardiac surgery patients [228,229,230,231]. The concentration of haptoglobin decreases with liver damage, which reduces serum anti-fHb potential [232]. A hemoglobin-dependent reduction in the NO pool occurs even under conditions of good serum iron-binding capacity, since the hemoglobin–haptoglobin complex binds NO similar to fHb [233]. 

Pathological manifestations depend on the blood level of hemoglobinemia and NO [234,235,236]. An increase in systemic and pulmonary pressure and the development of dystonia of abdominal smooth muscle cells lead to dysphagia, esophageal spasms and impaired intestinal microcirculation [237,238,239,240,241,242]. The sequestration of endogenous NO mediated by reducing the concentration of cGMP enhances the properties of platelets; moreover, fHb can lead to the activation of blood platelets in microvasculature spasms [221]. The microcirculatory distress formed from hemoglobinemia can be aggravated by thromboses with the circulation blockage and organ injury [243,244]. 

In regard to cardiac surgical patients at risk of developing AKI, studies show that the perioperative delivery of NO in cardiac surgical patients is safe. There is no significant increase in the inspiratory concentration of NO_2_, methemoglobin, hemodynamic disorders, cytokine changes or postoperative bleeding [245]. The first clinical work to study NO intervention at 80 ppm during and after CPB showed a decrease in AKI incidence and an improved 1-year kidney outcome in cardiac surgery patients [246]. Another cardiac surgical cohort showed that 40 ppm of NO administration to the CPB circuit leads to a decrease in AKI incidence and AKI severity, as well as improved kidney function [245]. The same group of authors demonstrated the intraoperative disturbance of NO homeostasis, assessed based on NO metabolites, which leads to the formation of perioperative NO deficiency in control patients [245]. Furthermore, a meta-analysis confirmed that perioperative NO delivery is associated with a reduced risk of AKI in cardiac surgery [247]. NO delivery does not reduce the risk of AKI when starting therapy after weaning from CPB. However, NO delivery significantly reduces the AKI risk when starting therapy at the initiation of CPB [247]. 

Thus, intraoperative hemolysis associated with cardiac interventions and CPB is an important factor in AKI, limiting the bioavailability of NO [153]. This fact is a powerful impetus for the development of new diagnostic and therapeutic strategies to improve patient outcomes in cardiac surgery. 

## 3. Translation of the Effects of Nitric Oxide into Clinical Practice

To date, the most extensive evidence base for the use of NO is in the pediatric patient cohort. The American Heart Association and American Thoracic Society recommend iNO to reduce the need for ECMO support in term and preterm infants with persistent neonatal PH or hypoxemic respiratory failure; iNO therapy can be used to improve oxygenation in infants with congenital diaphragmatic hernia and severe PH but should be used with caution in patients with suspected LV dysfunction [248]. iNO treatment can also be effective in infants with established bronchopulmonary dysplasia and symptomatic PH [248] iNO is mainly used in the ICU setting and is useful in patients with acute pulmonary vascular crisis and/or acute exacerbation of PH in the setting of an underlying parenchymal lung disease; iNO may be considered for the treatment of post-operative PH in mechanically ventilated patients to improve oxygenation and reduce the risk of pulmonary hypertensive crisis [249]. iNO, in addition to conventional postoperative care, should be used as the initial therapy for pediatric heart disease and failure of the right side of the heart [248]. iNO is used to perform an acute vasoreactivity test to determine the operability of heart disease and heart transplant candidates [250]. Transient changes in pulmonary vascular tone may contribute to right ventricular dysfunction or hypoxemia in adult cardiac surgery; NO use is associated with improved right ventricular function and increased cardiac output, decreased pulmonary vascular resistance, decreased pulmonary artery pressure and optimized ventilation-perfusion matching [251]. The pharmacological effects of NO are applicable to the treatment of hypoxia, right ventricular dysfunction, pulmonary hypertension and graft dysfunction in adult patients who underwent cardiac surgery [252]. iNO is widely used to treat pulmonary hypertension, right ventricular failure and hypoxemia in adult cardiac surgery patients [253]. The administration of iNO improved the ability of weaning from cardiopulmonary bypass, decreased the need for inotropic or vasopressor support, decreased the duration of postoperative mechanical ventilation and decreased intensive care unit length of stay during cardiac operations [254]. The use of iNO in heart transplant recipients leads to a decrease in pulmonary artery pressure and improvements in hemodynamic parameters, but does not affect long-term survival [255,256]. There is also little evidence to support the long-term benefit of iNO in lung transplant patients; however, some professional communities rate the efficacy of iNO as reasonable and recommend public health insurance for inhaled NO therapy during lung transplant [257]. The therapeutic potential of iNO, combined with a good safety profile, justifies its use despite the lack of definitive evidence to support improved long-term outcomes or survival among cardiac surgery patients, heart transplant patients, lung transplant patients and patients undergoing LVAD implantation [258]. Therefore, despite almost 30 years of application history, we still do not have a solid evidence base and clear recommendations for the use of NO in cardiac surgery [259]. There is insufficient evidence to reach a consensus on the dosage, duration of therapy or impact on patient outcomes for conventional NO indications [260]. Relatively recent clinical data on the effectiveness of NO in the framework of perioperative organ-protection strategies in cardiac surgery lead the clinician and scientist to a real “terra incognita” [261]. Probably, before the formation of a scientifically based view and clear practical recommendations, as well as translation into routine clinical practice, there is still a long way to go. At the moment, this is a rather promising area for future research.

## 4. Limitations and Future Directions

Despite the huge fundamental base and wide experience in the daily use of NO, its routine use for organ protection in cardiac surgery remains limited. The available pool of clinical trials shows a clear positive tendency in the use of NO, but does not allow us to draw unambiguous conclusions about its effectiveness in dramatically improving outcomes. This may be due to complex factors; in particular, no study has identified a target cohort of patients for perioperative NO organ protection. The optimal modality of NO treatment is still unknown, and the mechanisms of effector action have not been elucidated. Further clinical studies should include optimizing the NO-administration protocol in order to establish a sufficient NO exposure time and determine the regimen needed to prevent multiorgan injury in cardiac surgery [262]. Ideally, one should strive for the personalized administration of NO according to the optimal schedule in situations that offer the greatest benefit with the possibility of bedside assessment of exposure. Logistical problems and the high cost of application when using traditional cylinder-based technology also limit the scientific research, which requires new technological solutions. Possible prospects for further research are presented in Table 4.

The use of NO within the framework of the concept considered in this paper can encounter limitations. First, adjuvant organoprotection with NO in cardiac surgery is off-label administration. This reduces the possibility of implementing the technology in routine clinical practice and dictates the need for extensive clinical trials, which can become a scientific justification for including NO treatment in guidelines and standards for perioperative management in cardiac surgery. 

Secondly, despite not being identical to the conditions of cardiac surgery, there is serious concern about the role of NO in the development of complications in specific categories of patients, especially for sepsis. These concerns are related to studies using experimental models of sepsis, which showed the generation of high levels of NO [263]. However, further studies have shown that systemic NO production in patients with sepsis and shock is either unchanged or can actually decrease [264]. There is evidence for reduced NO bioavailability in human sepsis; moreover, the administration of NO via inhalation to patients with sepsis has been found to be safe [265]. The role of the NO pathway in the progressive organ damage during sepsis needs re-examination [266].

Thirdly, logistical problems and the high cost of NO therapy can be serious impediments [267]. A possible solution could be the development of new iNO-delivery devices that offer on-demand NO generation. There are various types of bedside NO synthesis and delivery technologies: electricity-generated NO systems, chemical-based NO systems, NO-releasing solutions and nanoparticle NO technology. These devices are at different stages of development: from pre- and clinical testing to regulatory approvals [268].

In addition to the administration of NO, under various scenarios requiring the use of organ-protective technologies, it is possible to use various NO donors and precursors. This is especially true in I/R-associated NO deficiency [269,270,271]. A promising prospect is the use of synthetic nanoparticles, which allow for targeted therapy and targeted delivery of NO to tissues damaged by ischemia and reperfusion [272,273].

## 5. Conclusions

Accumulating data suggest that the use of NO during cardiac surgeries with CPB provides myocardial and kidney protection. NO side effects are known, predictable, reversible and rare. The use of a native NO molecule does not affect systemic hemodynamics, which may be preferable compared to the use of pharmacological precursors, including nitrates. Postoperative vasoplegia, the development of systemic inflammatory response syndrome and sepsis can be contraindications to the use of NO. Deep physiological research is needed for a fundamental rationale, as well as the development of a new technological support for NO treatment. Further research is needed to determine the effect of NO delivery to prevent multiorgan injury and improve clinical outcomes in cardiac surgery patients. 

## Figures and Tables

**Figure 1 biomedicines-11-01085-f001:**
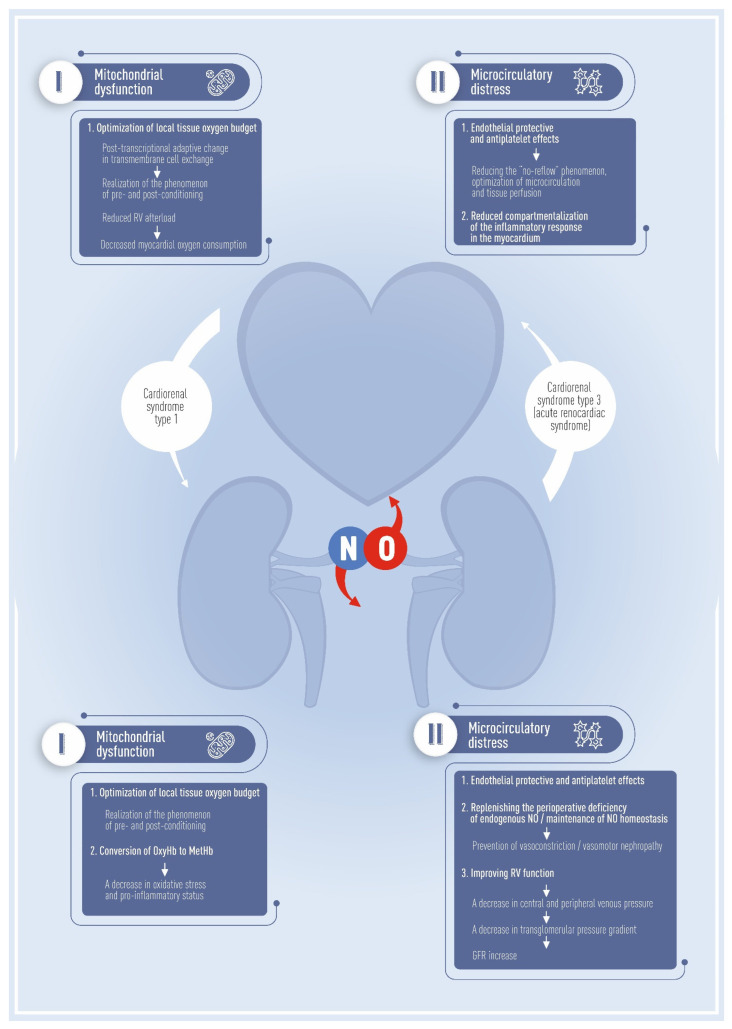
Nitric oxide in perioperative organ protection. The modern paradigm of the pathogenesis of perioperative myocardial and kidney injuries in cardiovascular surgery implies a close relationship of organ complications in the heart–kidney axis. The additive potentiation of multiple organ dysfunction is referred to as the cardiorenal continuum and includes both the direct association of cardiac dysfunction with the development of AKI (cardiorenal syndrome type 1) and reversible myocardial injury in the manifestation of AKI (cardiorenal syndrome type 3-acute renocardiac syndrome). The figure illustrates the fundamental basis for targeted perioperative prevention of organ dysfunction. NO therapy may prevent the cardiorenal continuum in cardiac surgery through a pluripotent positive effect on mitochondrial and microcirculatory distress in the heart–kidney axis. NB: Right ventricle (RV); oxyhemoglobin (OxyHb); methemoglobin (MetHb); glomerular filtration rate (GFR).

**Table 1 biomedicines-11-01085-t001:** Nitric oxide (NO) in the implementation of early and late preconditioning phenomena. A comparison of the NO role in early and late preconditioning.

	Early Preconditioning	Late Preconditioning
NO role	Trigger	Trigger and mediator
Implementation time	5 min	24 h
Duration of effect	1–2 h	48 h
Signaling isoforms of NOS	eNOS	iNOS eNOS
Second messengers	cGMP-independent mechanism	cGMP-dependent mechanism
Consequences at the cellular level	Optimization of bioenergetic activity, ion exchange and metabolism	Transcriptional activation, de novo protein synthesis, post-transcriptional modifications
Consequences at the tissue/organ level	Emergency adaptation	Phenotypic reprogramming
Physiological effects	-Antiarrhythmic -Cytoprotective -Endothelium-protective	-Cytoprotective -Prevention of development and reduction in the severity of stunning -Accelerated reversal of post-ischemic dysfunction -Antiarrhythmic -Antiapoptotic -Endothelium-protective
Clinical effects	-Increased threshold for arrhythmias -Reduction of infarct size in the I/R zone	-Reduction of infarct size in the I/R zone -Preservation of contractile function -Increased threshold for arrhythmias

NB: Nitric oxide (NO); cyclic guanosine monophosphate (cGMP); endothelial nitric oxide synthase (eNOS); inducible nitric oxide synthase (iNOS); ischemic/reperfusion (I/R).

**Table 2 biomedicines-11-01085-t002:** Mechanisms of NO-mediated cardiac protection.

Most Probable	Less Probable
Ca^2+^ incoming flow inhibition and limiting calcium overload Antagonism with B-adrenergic stimulation Contractility reduction Opening of sarcoplasmic and mitochondrial KATP tubules: Reduced myocardial consumption of O_2_ and glucose due to respiratory chain depression Preservation of ATP and the adenine nucleotide pool Antioxidant action Activation of COX-2, synthesis of cytoprotective prostaglandins	Preservation of endothelium-dependent coronary vasodilation Production of “signal” amount of ROS with limitation of oxidative stress reduction to the “no reflow” phenomenon Reduction of inflammation compartmentalization in the myocardium Leukocyte infiltration reduction Reduction in cytokine excretion and adhesion molecule expression

NB: ATP-sensitive potassium channels (KATP); adenosine triphosphate (ATP); cyclooxygenase 2 (COX-2); reactive oxygen species (ROS).

**Table 3 biomedicines-11-01085-t003:** Clinical studies using low-dose iNO (<80 ppm). Studies of cardioprotective effects of NO in different categories of patients in terms of cardiac surgery.

Authors	Design	Population	NO-Treatment Modality	Compared	Results in NO Group	Comments
Kolcz et al., 2022 [157]	A single-center RCT	97 children; 1.8 to 3.3 years; Fontan surgery	20 ppm; CPB period	-clinical outcomes -hemodynamics -myocardial injury -pro-and antiinflammatory mediators -metabolic stress	Shorter duration: -MV -ICU-stay -effusions Lower levels: -central venous pressure -cTnI, CPK-MB -lactate -glucose -CAI -IL-1 β, IL-6 and IL-8 Higher levels: -IL-10	extremely PVR-dependent population
Schlapbach L.J. et al., 2022 [158]	A multicenter RCT	1371 children younger than 2 years; different pathologies	20 ppm; CPB period	-clinical outcomes	Not different in VFD, LCOS, ECLS, death and ICU-stay	-an extremely heterogeneous group of patients in terms of severity and complexity of the pathology
Elzein C. et al., 2020 [159]	A single-center RCT	24 newborns; Norwood procedure	40 ppm; CPB period	-myocardial injury -clinical outcomes	Lower levels: -cTnI No differences: -VIS, duration MV, ICU-stay	-a small number of patients -nontypical CPB technique -postoperative routine use of iNO in both groups
Sardo S. et al., 2018 [160]	A meta-nalysis	958 adults and children	Different modalities	-clinical outcomes	Negligible reduction in ICU stay and duration of MV	-NO treatment only as a syndromic treatment of PH -large variability in dose, timing of the initiation, duration of NO treatment
Kamenshchikov N.O. et al., 2018 [161]	A single-center RCT	60 adults; CABG	40 ppm; CPB period	-myocardial injury -LV function	Lower levels: -cTnI, CPK-MB, VIS	-Short period of follow-up -mechanism of action was not studied
James C. et al., 2016 [162]	A single-center RCT	101 children; Different pathologies	20 ppm; CPB period	-clinical outcomes	Reduced the incidence of LCOS Decreased need for iNO in ICU -Decreased duration of ICU stay	-positive effects depend on age group and surgery complexity -groups of patients were not defined, operative procedures might be the most beneficial
Checchia P. et al., 2013 [163]	A single-center RCT	16 children; Tetralogy of Fallot	20 ppm; CPB period	-myocardial injury -LV function -clinical outcomes	Lower levels: -cTnI, BNP; Shorter duration: -MV, ICU stay Decreased need for diuretics	-mechanism of action was not studied -all patients received systemic steroids
Potapov E. et al., 2011 [164]	A single-center RCT	150 adults; after LVAD	40 ppm; After CPB +ICU	-RV function -clinical outcomes	No difference in RVD frequencies Tendency towards: -decrease in duration of MV; -needs for RVAD	-NO treatment only as a syndromic treatment to reduce the PVR -26.0% of the control group were switched to open-label iNO
Gianetti J. et al., 2004 [165]	A single-center RCT	29 adults; AVR+CABG	20ppm; OR+CPB+ICU	-myocardial injury -LV function -systemic inflammation	Lower levels: -cTnI, BNP, CPK-MB and P-selectin	-lacking major clinical end points

NB: Randomized clinical trial (RCT); aortic valve replacement (AVR); coronary artery bypass surgery (CABG); left ventricular assist devices (LVAD); operating room (OR); cardiopulmonary bypass (CPB); intensive care unit (ICU); left ventricle (LV); right ventricle (RV); cardiac troponin I (cTnI); brain natriuretic peptide (BNP); MB fraction of creatine phosphokinase (CPK-MB); right ventricular dysfunction (RVD); mechanical ventilation (MV); right ventricular assist devices (RVAD); pulmonary vascular resistant (PVR); vasopressor-inotropic scale (VIS); low cardiac syndrome output (LCOS); inhaled nitric oxide (iNO); ventilator-free days (VFD); extracorporeal life support (ECLS); pulmonary hypertension (PH); central venous pressure (CVP); pro-inflammatory interleukins (IL-1 β, IL-6 and IL-8); anti-inflammatory interleukins (IL-10); catecholamine index (CAI).

**Table 4 biomedicines-11-01085-t004:** Future directions. Prospects for further research investigating the beneficial effects of NO in cardiac surgery.

Scientific Areas	Exploratory Points
Determination of the target population	Comorbidity: Conditions and diseases predisposing to endogenous NO deficiency (AH, diabetes, multifocal atherosclerosis) Type of operation: Combined intervention, emergency operation, intraoperative blood loss and massive transfusions High risk patients: LV hypertrophy, incomplete revascularization, decreased LV EF, recent MI Determination of the basic NO status (concentration of metabolites in the blood, exNO)
Optimal modality	Determining the optimal initiation time, duration, dose of NO treatment Identification of time course of NO by-products and dynamics of exposure to NO metabolites
Search for mechanisms of organoprotective action	Determination of organ-specific effects and protective cellular mechanisms Identification of remote effects NO pathways NO and other organoprotective technologies: Implementation of the recommendation of KDIGO, GDP, et al.
Technological support	Production of bedside NO synthase and delivery technologies: Electricity-generated NO systems, chemical-based NO systems Working out methods for bedside performance monitoring

NB: Left ventricular (LV); nitric oxide (NO); arterial hypertension (AH); left ventricular ejection fraction (LV EF); myocardial infarction (MI); GDP (goal directed perfusion); Clinical Practice Guideline for Kidney Disease Improving Global Outcomes (KDIGO); exhaled NO (exNO).

## Data Availability

The datasets used and/or analyzed during the current study are available from the corresponding author on reasonable request.

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
