# Peer review of "Nitric Oxide in Cardiac Surgery: A Review Article"

_biomedicines, 2023, doi:10.3390/biomedicines11041085_

Round 1
Reviewer 1 Report
Kamenshchikov et al. have carried out an extensive and in-depth bibliographic review that deals with the role of NO in cardiac surgery and all the possibilities derived from this knowledge to prevent the development of sequelae in the patient after surgery.
Overall, it is a very complete review, well written and developed, and extensively documented with over 200 bibliographical references.
However, I think that some minor aspects should be resolved before its final publication:
- I think that two main headings should be added for sections 1 and 2 (the text starts directly with point 1.1. without introducing the topic in point 1; and the same happens with point 2).
- There are some writing errors (I recommend reviewing the text in depth). Some examples:
* Line 27: "С. ARDIAC" appears instead of "CARDIAC"
* Line 555: "a1available" appears instead of "available"
- Figure 1 should have better resolution. It's hard to read the text on it.
- Table 2 is not very understandable according to the format it presents. I propose that they modify it so that the hierarchies of the different phrases and titles are better understood. I think the same happens with Table 4.
- Bearing in mind that many abbreviations are used in the text and that, furthermore, it is a long text, I suggest including an abbreviation index at the beginning or end of the article.
Author Response
Dear Reviewer,
Thank you for the recommended revisions to Manuscript ID: biomedicines-2208012 entitled “Nitric oxide in cardiac surgery: a review article” by Nikolay O. Kamenshchikov, Nicolette Duong and Lorenzo Berra. A document has been included to state the your comments as well as the manuscript updates and response to each comment. The manuscript has been modified with track changes.

Reviewer 2 Report
The manuscript prepared by Kamenshchikov et al. focuses on the potential role of NO in cardiac surgery. During IRI, the nitric oxide (NO) levels restoration may alleviate reperfusion injury in ischemic organs such as the heart and kidney. The protective mechanism of NO is due to anti-inflammatory effects, antioxidant effects, and the regulation of cell signalling pathways.
To improve de quality of the manuscript, I recommended to the authors:
Major comments:
- Kindly add an Introduction section to the main text, which describes the current state-of-the-art view regarding the potential role of NO in cardiac surgery.
- Although it is a systematic review, please reconsider the references list to include new studies focusing on the role of NO in cardiac surgery (the title of the manuscript). From a total of 225 references, only 19th references from 2016 to 2022, and 72nd are older than 2000.
- The authors should improve the quality of Figure 1. It is hard to read the text in this Figure.
- Chapter 1.2 is incompletely described. Mitochondria have been suggested to be central targets by which NO can improve cardiac IR injury by opening the KATP channels that reduce Ca2+ overload. Supplementary, it prevents cytochrome c release and apoptosis. These aspects are incompletely described. NO appears to regulate inflammatory responses in ischemia-related injury. NO has a protective role in IR by downregulation of the MIP expression via MAPK (ERK1/2) and NF-κB pathways signalling. NO has an antioxidant effect and decreases reactive oxygen species (ROS) generation. Please see the review article Lundberg J and Eddie Weitzberg, E., 2022, DOI: 10.1016/j.cell.2022.06.010, Lee at al., 2022 https://doi.org/10.3390/antiox11010057 and Paul J. Fadel, 2017, https://doi.org/10.1161/HYPERTENSIONAHA.117.08999. In my opinion, this subchapter is a historical perspective of the NO in cardiovascular pathology.
- A more detailed description regarding the NO synthesis and cellular mechanisms in cardiac physiology (maybe in a new subchapter) could be a good factor in writing the same.
- Lines 192-194 should be in the Introduction section ’’This review summarizes studies which indicate that the implementation of the early PC phase cardioprotective effects induced by ischemia does not necessarily involve endogenous NO’’.
- In Table 1 - Please add a column with references according to the data presented.
- Kindly replace tables 2 and 4 with Figures to highlight the data presented.
- The authors should be considered to add in subchapter 3. (LIMITATIONS AND FUTURE DIRECTIONS) a description of the new therapeutic approach in cardiac IR, from direct NO administration to different NO Delivery Systems (please see Lee et al., 2022 https://doi.org/10.3390/antiox11010057)
Minor comment:
- Please add references related to data presented in lines: 37, 45, 98, 102, 137, 153, 165, 171, 191, 199, 254, 258, 281, and 303.
- Kindly list the studies in the decreasing order of the year appearance in Table 3
Author Response
Dear Reviewer,
Thank you for the recommended revisions to Manuscript ID: biomedicines-2208012 entitled “Nitric oxide in cardiac surgery: a review article” by Nikolay O. Kamenshchikov, Nicolette Duong and Lorenzo Berra. A document has been included to state your comments as well as the manuscript updates and response to each comment. The manuscript has been modified with track changes.

Round 2
Reviewer 2 Report
Thank you for allowing me another opportunity to revise this manuscript. Many suggestions regarding the structure of the initial manuscript have been included.
My concern remains regarding the quality of figure 1. The authors should include the revised figure 1 in the manuscript.
In addition, please revise the reference list according to the Biomedicines Journal recommendation.
Author Response
Dear Reviewer,
Thank you for the recommended Round 2 revisions to Manuscript ID: biomedicines-2208012 entitled “Nitric oxide in cardiac surgery: a review article” by Nikolay O. Kamenshchikov, Nicolette Duong and Lorenzo Berra.
According to your suggestions:
- We included the revised Figure 1 in the manuscript and also improved its quality (increased resolution and changed the background so that the letters on it would look more contrast and readable)
- We revised the reference list according to the Biomedicines Journal recommendation